# Priority Actions to Advance Population Sodium Reduction

**DOI:** 10.3390/nu12092543

**Published:** 2020-08-22

**Authors:** Nicole Ide, Adefunke Ajenikoko, Lindsay Steele, Jennifer Cohn, Christine J. Curtis, Thomas R. Frieden, Laura K. Cobb

**Affiliations:** Resolve to Save Lives, Vital Strategies, New York, NY 10005 USA; AAjenikoko@resolvetosavelives.org (A.A.); lsteele@resolvetosavelives.org (L.S.); jcohn@resolvetosavelives.org (J.C.); cjohnson@vitalstrategies.org (C.J.C.); trfrieden@resolvetosavelives.org (T.R.F.); lcobb@resolvetosavelives.org (L.K.C.)

**Keywords:** sodium, salt, sodium reduction, cardiovascular disease, blood pressure, hypertension, nutrition, health policy

## Abstract

High sodium intake is estimated to cause approximately 3 million deaths per year worldwide. The estimated average sodium intake of 3.95 g/day far exceeds the recommended intake. Population sodium reduction should be a global priority, while simultaneously ensuring universal salt iodization. This article identifies high priority strategies that address major sources of sodium: added to packaged food, added to food consumed outside the home, and added in the home. To be included, strategies needed to be scalable and sustainable, have large benefit, and applicable to one of four measures of effectiveness: (1) Rigorously evaluated with demonstrated success in reducing sodium; (2) suggestive evidence from lower quality evaluations or modeling; (3) rigorous evaluations of similar interventions not specifically for sodium reduction; or (4) an innovative approach for sources of sodium that are not sufficiently addressed by an existing strategy. We identified seven priority interventions. Four target packaged food: front-of-pack labeling, packaged food reformulation targets, regulating food marketing to children, and taxes on high sodium foods. One targets food consumed outside the home: food procurement policies for public institutions. Two target sodium added at home: mass media campaigns and population uptake of low-sodium salt. In conclusion, governments have many tools to save lives by reducing population sodium intake.

## 1. Introduction

Excess consumption of dietary sodium increases blood pressure and therefore the risk of cardiovascular disease (CVD) [1,2,3]. In 2017, high sodium intake led to approximately 3 million deaths and loss of 70 million disability-adjusted life-years [4]. These devastating outcomes affect a wide range of populations, with four out of five of these deaths occurring in low- and middle-income countries (LMICs), and nearly half among people younger than 70 [5]. Dietary interventions, including limiting sodium consumption from all sources, are an important approach to addressing key risk factors for non-communicable diseases [6].

The World Health Organization (WHO) recommends adults consume less than 2 g of sodium (5 g salt) per day. However, in 181 of 187 countries, estimated average levels of sodium intake exceed recommendations, with the global average sodium consumption estimated to be approximately 3.95 g per day [5].

Dietary sodium is typically consumed from three distinct sources: sodium added to packaged food during manufacturing, sodium added to food consumed outside the home (e.g., restaurants, street food vendors, cafeterias/canteens), and sodium added in the home, either during cooking or while eating (also known as discretionary sources). The primary source of sodium intake varies widely by region and country. Many Asian countries as well as a majority of LMICs that have documented sodium sources consume a majority of their sodium from discretionary sources. Others, including the United States, the United Kingdom and Australia, consume sodium primarily from processed food sources [7]. For an example of these differences, refer to Figure 1.

### 1.1. Action to Date on Sodium Reduction

Efforts to reduce sodium have been increasing at national, regional, and global levels. The number of countries that have national sodium reduction strategies doubled from 2010 to 2014 [11]. As of 2016–2017, 89 countries have national policies that support sodium reduction [12]. Although the specific strategies vary, almost all take a multi-component approach. Consumer education is the most common intervention, although an increasing number incorporate a regulatory approach as well. These are either sodium-specific, such as setting sodium targets for packaged food, or part of addressing broader nutrition goals, such as front-of-pack labeling [11].

Globally, Member States at the 66th World Health Assembly in 2013 united by setting a voluntary global target of a 30% relative reduction in mean population sodium intake by 2025. The WHO then released “SHAKE the salt habit: The WHO’s SHAKE technical package for salt reduction,” which assists Member States with sodium reduction strategies. Regional efforts have helped to spur action, for example with the Pan American Health Organization and the European Salt Action Network organizing on sodium reduction to stimulate country action and provide technical support. Further, civil society is working to generate progress on sodium reduction, including the World Action on Salt and Health network, with members from 100 countries.

As countries progress on sodium reduction, some concern has been raised that these efforts could lead to inadequate intake of iodine, given that many populations primarily consume iodine through fortification of salt. Iodine fortification has been a major and successful public health initiative over the past two to three decades. Thus, it is important that progress in this area is not hindered but complemented by salt reduction strategies. Iodine fortification and salt reduction programs can be integrated to optimize intake of both nutrients by integrated surveillance of salt and iodine intake [13,14].

### 1.2. Evidence for Sodium Reduction Interventions

Sodium reduction is a high-impact and cost-effective strategy to improve health. WHO recommends multiple sodium reduction interventions as “best buys,” including reformulation of food products, establishing a supportive environment in public institutions by providing low-sodium options, education through behavior change communication and mass-media campaigns, and front-of-pack labeling. WHO estimates a US$13 return on every US$1 invested in these interventions [15]. Systematic reviews of the impact of sodium reduction interventions conclude that multi-component, population-level policy interventions that create structural changes to the environment in which people make decisions (e.g., mandatory reformulation) generally have a larger impact on reducing sodium consumption than strategies targeting behavior change alone (e.g., dietary counseling) [16,17,18]. The framework we apply here builds on published reviews and experience during implementation to provide additional guidance for governments and others to identify priority interventions and take systematic action to reduce sodium intake.

National salt reduction programs have demonstrated success in lowering population sodium intake as well as improved health outcomes. For example, both the United Kingdom and South Korea’s salt reduction programs achieved major reductions in population sodium intake (15% in UK adults and 24% in South Korean adults) through multi-component programs that included at least one intervention to address each of the three sources of sodium (see Table 1 for program details). In both countries, simultaneous decreases in adverse outcomes were observed, including decreased average systolic blood pressure and decreased deaths from stroke and ischemic heart disease in the United Kingdom, and, in South Korea, decreased blood pressure and hypertension prevalence [19,20,21].

In this article, we propose a framework for prioritizing interventions that have the highest potential to reduce population sodium intake, evaluate existing interventions against the proposed framework, and identify and describe the recommended high-impact interventions.

## 2. Framework Development

### 2.1. Existing Sodium Reduction Strategies

Through a review of scientific literature, WHO global nutrition reports, and online nutrition databases, we compiled a list of the major sodium reduction strategies used by countries around the world [11,12,22]. Table 2 below summarizes the sodium reduction strategies identified through our review. We organized the strategies based on the primary source of sodium that the strategy addresses.

### 2.2. Developing a Framework for Prioritizing High-Impact Sodium Reduction Strategies

Given the extensive range of existing sodium reduction strategies, countries must be able to prioritize the most effective interventions, as not all strategies have the same ability to achieve large-scale population sodium reduction. As noted above, structural or policy interventions are more effective than those that rely solely on individual behavior change [16,17]. Although this can help prioritize interventions, not all sources of sodium are equally amenable to policy solutions. Considering the many countries whose primary source of sodium comes from home-cooked foods, it is essential to identify a suite of strategies effective for all sources of sodium.

We propose a framework that aims to identify the highest-impact strategies that suit countries’ population sodium consumption profiles. The framework, described in detail in Box 1 below, specifies three inclusion criteria to consider. First, the strategy should be scalable, that is, able to be implemented at scale within a population in order to have sustainable benefits. Second, the strategy should have evidence of effectiveness, or, if there is no strategy proven effective for a particular source of sodium, be an innovative and promising approach. Third, the strategy should have a large benefit with the potential to reduce sodium consumption substantially at the population level.

Box 1A framework for prioritizing sodium reduction strategies.1.
**Scalable and Sustainable**
The strategy is able to be implemented at scale within a population rather than relying on individual-level interventions. It is feasible to support the inputs and costs necessary to sustain the effects of the intervention.
*Example: Chile’s Law of Food Labeling and Advertising [23]*
2.
**Evidence of effectiveness or innovation**
The strategy should meet at least one of the following criteria: (a)Rigorously evaluated with demonstrated success in sodium reduction
*   Example: United Kingdom’s food reformulation targets for packaged food [19]*
 (b)Suggestive evidence of effectiveness in reducing sodium consumption from lower quality evaluations or modeling studies
*   Example: Hungary’s Public Health Product tax on applicable salty snacks and condiments [24]*
 (c)Rigorously evaluated strategies that do not specifically target sodium reduction but could be adapted to address sodium and then evaluated to assess impact on sodium intake
*   Example: Mass media campaigns for tobacco control [25]*
 (d)Innovation: If a specific source of sodium in a population cannot be sufficiently addressed by a strategy meeting one of the 3 sub-criteria above, a strategy taking an innovative and promising approach may be included with a planned evaluation
*   Example: Providing subsidies for low-sodium salt in the Salt Substitute and Stroke Study in China [26]*
3.
**Large benefit**
The strategy is worth the investment: it is able to produce substantial, population-level reductions in sodium consumption or improvements in targeted health outcomes (e.g., reductions in blood pressure or CVD events), demonstrated by either (1) impact evaluations showing at least a 10% reduction in the sodium consumption in the target population, (2) scientific modeling of the strategy and its potential impact on sodium consumption and/or health outcomes, or (3) a theoretically large impact based on theory of change with significant effects on the primary source of sodium.
*Example: Modeled outcomes resulting from 10-year graduated sodium reductions in the United States packaged food supply suggest a 22% reduction in people with systolic blood pressure ≥140 mmHg, 895,000 CVD events, and 252,500 CVD-related deaths over 10 years [27].*


An individual country engaged in developing a suite of strategies for sodium reduction can prioritize strategies meeting the three criteria above based on their primary sources of sodium. Thus, the suite of high-priority sodium reduction strategies may vary between countries consuming sodium primarily from discretionary sources versus countries consuming large quantities of packaged or restaurant foods.

## 3. Applying the Framework

### 3.1. Evaluate Existing Interventions against the Proposed Framework

We reviewed the existing strategies described in Table 2 to determine which would meet the framework criteria of being scalable and having evidence of effectiveness or innovation. Seven strategies were found to meet all three framework criteria, which are listed in Table 3 along with a justification for inclusion.

### 3.2. Summaries of High-Impact Recommended Strategies

In the section below, each of the seven recommended strategies for population sodium reduction is summarized, along with examples of successful implementation. The interventions are organized by the primary source of sodium that the strategy addresses, although in some cases the strategy may affect more than one sodium source. For example, low-sodium salt substitutes primarily are used to reduce sodium in home-cooked foods, but they could also be used to reduce sodium in packaged foods or food prepared outside the home such as in restaurants or canteens. These summaries are intended to help country governments and other organizations identify and prioritize the highest impact strategies for population sodium reduction.

#### 3.2.1. Sodium Reduction Strategies for Packaged Food

The following four strategies affect packaged food either by directly altering the food manufacturing process or by regulating the price or promotion of packaged food items based on the content. Strategies for packaged food that were not included primarily consisted of labeling strategies that have not been proven to be highly effective, such as back of package nutrient declarations or regulating health claims on packaging, as well as supermarket interventions which are not supported with evidence of effectiveness or impact [28,29].

##### Front-of-Pack Labeling Regulations

Front-of-pack food labels can empower consumers to make healthier choices by using simple graphics to quickly convey information about the healthfulness of products [30]. These labels often address multiple unhealthy nutrients, including sodium, sugars, and saturated fats and/or trans fat. Although many countries require nutrition information on the back of packages, these labels are often hard for consumers to understand and do not lead to the selection of healthier options [22,31]. Overall, negative labels such as the “high in” warning labels have the most potential to decrease demand for unhealthy packaged foods more generally [32], and are the most effective labeling system to date. A strong, clear front-of-pack labeling system can encourage and support industry reformulation and be used to guide implementation of other policies such as public food procurement standards or marketing restrictions.

Front-of-pack labels vary in their objectives and style. Nutrient-specific systems use graphics to indicate to consumers the levels of unhealthy nutrients (e.g., “high in” warning labels; multiple traffic lights). Summary indicator systems combine several criteria to establish a scale for the healthfulness of a product (e.g., Nutri-score, star rating systems). Although indicator systems are more effective in helping consumers identify the healthiest products, others such as warning labels work by attracting consumers’ attention, improving understanding of nutrient content, reducing the perceived healthfulness of unhealthy products, and changing purchasing intentions [32,33]. Warning labels also require less processing time to understand and prioritize the information, even compared to other interpretive labels such as traffic light labels [34]. Key elements of an effective label include a prominent display and use of colors or graphics, being simple and easy to understand and use, making a judgment about the nutrition quality of the food (healthy or unhealthy) that consumers can easily understand, establishing thresholds for harmful nutrients, and being mandatory.

Front-of-pack labeling regulations are a scalable approach with suggestive evidence of effectiveness from multiple countries. This strategy has gained momentum globally in the past decade and is now being implemented in thirty countries [22]. There has been an increasing trend toward mandatory systems: labels in Mexico, Croatia, Chile, Uruguay, Peru, Israel, Iran, and Sri Lanka are all mandatory. Evaluations of implemented front-of-pack labeling systems show a positive effect on consumers’ purchasing behaviors and industry’s formulation of packaged foods [35]. For example, six months after Chile’s implementation of “high in” warning labels, there was an impact on both consumer purchasing behavior and industry reformulation of food products [36]. Results on the impact of Chile’s law on sodium-specific outcomes are still pending; however, results have shown that the purchase volume of “high-in” beverages decreased by 24% since implementation. Similarly, one year after Ecuador mandated traffic light labeling, consumers were choosing products with “medium” or “low” levels of unhealthy nutrients rather than “high” levels [37].

##### Food Reformulation Targets for Packaged Food (Voluntary or Mandatory)

Promoting industry reformulation through government-developed sodium targets for categories of packaged food creates change in the food environment without requiring consumer action or even knowledge. Sodium targets can be developed as a mandatory regulation or voluntary program, and ideally should target processed food categories that contribute most to sodium intake. Although voluntary approaches can be initiated more quickly in countries and are relatively flexible, mandatory approaches can level the playing field across food processing sectors and potentially have a larger impact [38]. A voluntary approach is dependent on active government leadership and strong commitment to monitor progress and hold industry accountable, commitment and action from the food industry to reach targets, and a potential for regulation if industry does not meet the targets. Consumer and civil-society advocacy can be important to enact and sustain these interventions, which are sometimes opposed by parts of the food industry. Whether voluntary or mandatory, most targets aim to reduce sodium in products by 20–25% over 5 years. Setting interim targets can allow for gradual sodium reduction over time and may be more acceptable and feasible for industry.

Strong evidence shows that both voluntary and mandatory approaches that take a comprehensive approach to target-setting across multiple food categories can be effective, leading to reduced sodium content in processed food. There is also evidence that target-setting can lead to meaningful reductions in sodium intake. The United Kingdom, which provides the strongest case study in setting voluntary targets, developed targets for 85 processed food categories in 2005. Adults’ salt intake decreased by approximately 15% between 2003 and 2011, with additional decreases in average blood pressure in the population and deaths from CVD [19]. However, progress reversed in 2011 when the Public Health Responsibility Deal was introduced, a public-private partnership that loosened government oversight of the targets and gave the food industry more freedom to self-monitor and self-regulate. Since 2011, annual declines in salt intake have significantly slowed [39].

Mandatory limits for sodium in at least one food category have been set in a variety of countries; however, targets set in only one or a limited number of categories may only have a large benefit if they address a category significantly contributing to population intake. South Africa and Argentina are to date the only countries that have set extensive mandatory regulations. Both countries passed legislation in 2013 setting maximum sodium levels for key food categories (18 types of products in 3 categories in Argentina and 13 categories in South Africa) [40,41]. Evaluations in each country have demonstrated compliance with the law, though the impact on sodium intake is not yet known. A 2019 evaluation in Argentina found that more than 90% of the products included in the law were found to be compliant [42], and a 2016 evaluation in South Africa found that at the time of implementation (2016), 67% of targeted foods were compliant with the legislated limit [43].

##### Regulation of Marketing of Foods and Nonalcoholic Beverages to Children

Food and beverage marketing that targets children overwhelmingly advertises unhealthy foods and is widespread and increasing with the growing reach of digital media [44]. Regulating food and beverage marketing to children, whether on food packaging, in schools or supermarkets, or through media, television, or the Internet, prevents frequent exposure to products high in sodium, fat, and sugar. Children are particularly vulnerable to food marketing as they are highly impressionable and typically unable to recognize advertising intent [45]. Food and beverage marketing adversely impacts children’s food and drink preferences, requests, and consumption [46]. International consensus supports implementing marketing restrictions that protect all children up to the age of 18, in line with governments’ duty to protect children’s right to health [46,47]. As early exposure to salty foods can determine salt preference later in life [48], limiting children’s exposure to these foods helps protect them from preferring a high salt diet as adults. This may also limit risk of developing high blood pressure later in life: a high-salt diet is positively associated with an age-related rise in blood pressure [3].

Marketing restrictions have most commonly been implemented as government-approved voluntary approaches and sometimes as self-regulatory approaches. However, mandatory regulations that include provisions for enforcement are more effective than self-regulated initiatives [22,49,50,51,52,53,54,55]. A small but growing number of countries have developed mandatory regulations to restrict unhealthy food marketing, with suggestive evidence of effectiveness. Chile’s Law of Nutritional Composition of Food and Advertising mandates restrictions on multiple forms of marketing, including child-targeted packaging and television advertising. Evaluations have documented significant decreases in child-directed advertising on packages [56] and the percentage of ads for regulated products decreased from 42% before the regulation to 15% after the regulation [57,58]. South Korea placed restrictions in 2010 on TV advertising of Energy-Dense and Nutrient-Poor (ENDP) foods targeting children, and as a result, the total advertising budget, advertisement placements, and gross rating points for ENDP foods decreased during regulated hours [59]. Although regulations are specifically targeted at marketing to children, they have the ability to change marketing to people of all ages as children often consume similar shows and media as adults. In Chile, a decrease in unhealthy food marketing occurred in programs intended for children by 37% as well as for programs for general audiences by 23% [58]. Using broad definitions of marketing to ensure inclusion of any media source to which children up to least age 16 are exposed can allow for maximum impact. Despite the evidence from these success stories, global progress is limited, and stricter measures are required to keep up with the food industry’s evolving marketing tactics.

Robust marketing restrictions are a scalable way to effectively reach an entire population. Although rigorous evaluations linking marketing restrictions to significant reductions in sodium consumption have not yet been conducted, there is evidence of the impact of unhealthy food marketing on increased consumption of foods high in sodium, sugar, and/or fat among children. Further evidence demonstrates that these restrictions can reduce all forms of marketing of high sodium foods, including advertising, product placement and branding, sponsorship, and product design and packaging. Marketing restrictions might also encourage manufacturers to reformulate their products to improve the healthfulness of their products, thus avoiding the regulations.

##### Fiscal Policies: Taxation on High Sodium Foods

Growing evidence shows that taxation can be a useful tool to discourage the consumption of unhealthy products. Taxes can also promote reformulation of products, benefitting all consumers [60]. Historically, excise taxes have most successfully been applied to reduce consumption of alcohol and tobacco [61,62]. More recently, taxes on sugar-sweetened beverages have become increasingly common, alongside a more limited number of taxes on products high in sodium, sugar, and/or fat (known as junk food taxes) [60,63]. There are multiple approaches to unhealthy food taxes; however, the key to success is that the price increases enough to discourage consumption and the tax is structured so that consumers substitute with products that have a healthier profile. In some cases, to reduce the burden on consumers and maintain demand, manufacturing companies choose to lower product prices to absorb some of the price increase. A combination of taxes along with subsidies for healthy foods (e.g., fresh fruits and vegetables) may reduce the burden that taxation may impose on lower-income people who do not change their purchasing patterns, and also improve healthy eating all around.

Strong evidence exists that sugar-sweetened beverage taxes are generally effective in reducing the sales and intake of sugar-sweetened beverages when taxes are substantial (e.g., at least 1 U.S. cent per ounce) [64]. For example, Mexico’s tax (1 peso per liter) led to an average reduction in sugar-sweetened beverage purchases of 7.6% during 2014–2015 [65]. There is less evidence for taxes on high-sodium products; however, the number of countries applying excise taxes to food products high in sodium is growing, including a tax on salty snacks in Mexico, salty snacks and condiments in Hungary, and instant noodles in Tonga [24,66,67]. Hungary’s Public Health Product Tax, introduced in 2011, taxes certain products high in sugar and salt, including salty snacks and condiments which exceed a maximum threshold level. Although an overall decrease in consumption of salty snacks or condiments has not yet been documented, 11–16% of consumers reported changing their habits due to the higher prices by either reducing their consumption or changing to a cheaper product, different kind of product, or different brand [24]. In Mexico, the 8% tax on non-essential foods, including salty snacks, resulted in a 6.3% decrease in total purchases compared to expected purchase rates [66].

Although real-world implementation of taxation on high-sodium foods is limited, modeling studies on the impact of sodium taxes predict decreased consumption and health gains [68,69,70,71]. These taxes primarily aim to reduce consumption of the taxed products; however they can also work to increase public awareness of the harms of these products, incentivize industry to reformulate their products and market healthier products, as well as generate new revenue that can be used to fund important health and nutrition programs [72,73,74,75]. Some research suggests that taxes focusing on a single nutrient, such as sodium, may have unintended consequences by increasing intakes of other unhealthy nutrients [76,77]. Incorporating multiple nutrients through a tax on unhealthy foods based on nutrient profiling (e.g., junk food taxes) may prevent these consequences.

#### 3.2.2. Strategies to Reduce Sodium Consumed from Food Prepared Outside the Home

Only one strategy in this category was identified that met the framework criteria. Interventions that target restaurant settings are not included. Restaurant interventions have been implemented in a few countries. Those that rely on in-person education of chefs/cooks are not scalable and sustainable. For others, such as eliminating salt shakers or placing warnings on high salt items in chain restaurants, there is little evidence that they have the potential to substantially impact population sodium consumption. Development of high-impact strategies for restaurants or street food is needed, particularly for countries with high sodium consumption from these sources.

##### Standards for Sodium as Part of Food Procurement Policies in Public Institutions

Public food procurement policies allow governments to influence the health of their populations by requiring that all food purchased by government and all food served or sold in government settings meets a specified set of healthy nutrition standards, including but not limited to standards for sodium. Although these policies are most often implemented in schools, they can also apply to food served or sold in public sector settings such as childcare centers, public hospitals and health centers, community centers, government offices, nursing homes, military bases, and prisons. These policies are typically implemented in government-funded institutions, but they can also extend to private sector organizations (e.g., private schools) and can be used to guide private companies in purchasing, serving, and selling healthy food. In addition, public food procurement can be structured to reinforce other recommended interventions, such as restricting food advertising in government settings and using key media campaign messages.

Public food procurement policies offer a scalable approach that can have a large reach. Meals and snacks served and sold by the government typically represent a large population. For example, in India, nearly 100 million beneficiaries received meals under the school Mid Day Meal Scheme from 2017–2018, and New York City’s nutrition standards impact an estimated 240 million meals and snacks served annually [78,79]. Public food procurement policies are a low-cost, high-impact way to improve diets and encourage reformulation without requiring direct industry regulation [80].

Suggestive evidence of effectiveness comes from a limited number of evaluations, mostly conducted in school settings. A few jurisdictions, including the United Kingdom, Singapore, U.S. cities such as New York City and Philadelphia, and the Australian Capital Territory have developed comprehensive, mandatory policies that cover a variety of public institutions such as schools, hospitals, and public sector offices [79,81,82,83,84,85,86]. Evaluations of these policies commonly show that public food procurement policies are effective at increasing the availability, purchase, and consumption of healthy food and decreasing purchase and consumption of unhealthy food. A more limited number of evaluations demonstrate further benefits such as reduction in blood pressure and/or body mass index [87,88,89].

#### 3.2.3. Strategies to Reduce Sodium Consumed at Home

Two strategies were included for reducing discretionary salt consumption. Because there was only one scalable and large-benefit strategy identified that was supported by evidence (mass media campaigns), we chose to include a second innovative strategy (increase uptake of low-sodium salt). Identifying multiple approaches was especially important for this category as many countries consume most sodium at home. Some interventions reviewed that address home consumption of sodium through other individual or behavior change approaches were found to successfully reduce sodium consumed at an individual level. However, these strategies have not been included, as they are difficult to scale and may not have a large benefit at the population level.

##### Mass Media Campaigns

Mass media campaigns can disseminate well-defined, behaviorally focused messages to large audiences via TV, radio, outdoor media, billboards, posters, and/or print media (i.e., magazines/newspapers). The campaigns may be a standalone intervention seeking to educate the public on a specific topic (e.g., dangers of high sodium consumption), or they may be part of a broader strategy to build support for or educate about the broader intervention (e.g., educating consumers on a recently enacted nutrition policy). Mass media campaigns often aim to invoke cognitive or emotional responses that affect decision-making processes at the individual level, and they may also stimulate interest and discussion regarding the campaign message.

Although multiple behavior change/educational approaches were originally identified that could drive behavior change on an individual level, the only strategy scalable to a large population was mass media campaigns. For decades, mass media campaigns have been used to reduce smoking and tobacco use. Strong evidence exists for these campaigns, with evaluations showing an association with decreased initiation among young people and an increase in cessation among adults [90]. Successful campaigns require adequate exposure to the campaign and often rely on being able to elicit negative emotions such as fear, disgust, and sadness [91,92].

Several mass media campaigns have focused on nutrition or physical activity, many with positive results such as increased consumption of fruits and vegetables or promoting walking [93,94,95,96]. Limited evidence exists for mass media campaigns contributing to reduced sodium consumption. In South Africa, an advocacy group called Salt Watch developed a mass media campaign in 2014 to increase awareness of the dangers of high sodium consumption, which included messages on reducing discretionary sodium intake. An assessment found that 75% of participants had seen the specific Salt Watch media campaign and significantly more participants who had seen the campaign reported taking steps to control their salt intake compared to those who had not seen it [97]. In the UK, a consumer awareness campaign using a variety of media outlets was conducted as part of the national salt reduction campaign. A study of this campaign found that the number using salt at the table declined significantly after the campaign, from 32.5% in 2003 to 23.2% in 2007 [98]. Mass media campaigns may be an effective tool to use as part of a strategy to reduce discretionary consumption of salt among the population along with other methods. However, maintaining the benefits of a campaign require sustained efforts, such as repeated broadcasting over time; thus, these campaigns will only have a large benefit if sufficient budget is available.

##### Increase Uptake of Low-Sodium Salt (Promotion, Distribution, Subsidies)

Low-sodium salt substitutes typically replace a portion of the sodium in the salt (usually 10–30%) with alternative minerals, most commonly potassium that has added benefits on blood pressure reduction. Like standard salt (NaCl), it can be fortified with iodine. Strong evidence from randomized trials shows that low-sodium salts reduce blood pressure, and more limited evidence indicates that it results in reduced CVD and mortality [99,100,101,102]. A modeling study that estimated the potential effects of nationwide replacement of discretionary salt with a potassium enriched salt substitute in China found that approximately 460,000 CVD deaths could be prevented each year, including 208,000 from stroke and 175,000 from heart disease [103].

Use of low-sodium salt in the home is not widespread in most countries. Studies in China have shown the major barriers to be lack of awareness and the higher cost of low-sodium salt [104]. These barriers can be removed through promotion of low-sodium salts, ensuring low-sodium salts are widely distributed and available in the market, and using subsidies to equalize the price of low-sodium salt and standard salt. Removing price as a barrier can increase demand for low-sodium salt among consumers and, potentially, among industry and restaurants as well. This increased demand would at minimum, encourage increased production, and could potentially lower production costs. Promotion of low-sodium salt has the risk of increasing overall salt consumption, which might increase potassium consumption but not decrease sodium consumption.

Increasing uptake of low-sodium salts through promotion, wider distribution, and/or subsidies is an innovative and potentially high-impact strategy for reducing high sodium intake, particularly for countries in which sodium consumed in the home is the primary source. Limited evidence from novel studies demonstrates the potential of these interventions. A trial in China showed that providing subsidies for low-sodium salt nearly doubled the reported use of the low-sodium salt [104]. A randomized controlled trial in Peru demonstrated that low-sodium salt is both acceptable in the population and effective in reducing blood pressure and hypertension incidence [105,106].

There is some concern that use of potassium-based salt substitutes for individuals with impaired potassium excretion could increase the potential for adverse effects due to hyperkalemia, including increased risk of arrhythmias and sudden cardiac death. However, scientific communities such as the Scientific Advisory Committee on Nutrition have concluded that “at a population level, the potential benefits of using potassium-based sodium replacers to help reduce sodium in foods outweigh the potential risks” [107]. Achieving target sodium levels for countries primarily consuming sodium in the home may not be feasible by relying on behavior change alone. Enacting policies that promote low-cost, low-sodium salt may be a more effective and sustainable approach.

## 4. Discussion

In this article, we describe seven potentially high-impact strategies for sodium reduction. By identifying the strategies that are scalable and have evidence of effectiveness or innovation, governments can concentrate on the highest-impact interventions that address excess sodium consumption. A majority of the interventions developed to date address sodium from packaged food, and we recommended four of these strategies: setting voluntary or mandatory reformulation targets for sodium in packaged food, front-of-pack labeling regulations, regulation of marketing of foods and nonalcoholic beverages to children, and taxation of high-sodium foods. Although a variety of interventions exist for reducing sodium in food prepared outside the home, our review only identified one strategy as high-impact: standards for sodium as part of food procurement policies for public institutions. Few strategies have been developed that reduce sodium consumption in the home. We propose that the highest-impact, most scalable strategies for this area include mass media campaigns and increasing uptake of low-sodium salt substitutes through promotion, distribution, and/or subsidization.

Strategies should be appropriate to the local context in terms of the sources of dietary sodium, existing regulatory mechanisms, and access to resources. No single strategy is enough to reach the WHO goal of a 30% reduction in sodium intake by 2025, thus a multi-component package is needed. This is especially true for populations that primarily consume sodium added in the home, where interventions have typically relied on educational approaches which have shown very limited benefit at a population scale. Nutrition education, including through a mass media campaign, is far more effective when combined with an environmental and policy approach [108,109].

Studies and pilot interventions that relied primarily on an educational/individual approach have reduced sodium consumption among study participants. For example, the Trials of Hypertension Prevention (TOHP) included comprehensive education and counselling on sodium reduction among prehypertensive adults in the United States. Results showed that participants in the intervention achieved significant reductions in sodium and a 25–30% lower risk of cardiovascular outcomes in the 10 to 15 years following the trial [110]. Another study from northern China involving a school-based salt reduction education program (School-EduSalt) found reduced salt intake and blood pressure among children and their family members in the intervention group [111]. Such examples demonstrate that with high levels of investment, individual-focused educational interventions can successfully increase motivation and knowledge leading to sodium reduction behaviors. However, these investments are likely cost-prohibitive to conduct at scale and difficult to sustain without a change in the food environment. A review of such educational studies warns that despite the possibility of success, they are unlikely to be adequate in reducing population salt intake to the recommended levels and would benefit from being implemented alongside more structural interventions [112].

Therefore, because of the difficulty of scaling these interventions and achieving population level benefits, we did not include community education or individual education and counseling as recommended standalone strategies. These education-based strategies are particularly challenging for sodium reduction, as there are typically no visible consequences to a high-sodium diet and sodium-related non-communicable diseases often have no symptoms until they have reached an advanced stage. Thus, many individuals lack the motivation to change, regardless of their level of awareness. Education can, however, be used to complement many of the included strategies and improve their chances of achieving large population benefits. For example, the effectiveness of front-of-pack labeling policies can be strengthened through a complementary education campaign on how to use/interpret the new labels [113,114]. Similarly, health care workers could be trained to promote low-sodium salt to their patients.

Some of the recommended strategies take an approach specific to sodium, such as setting reformulation targets for packaged food and increasing uptake of low-sodium salt, while other strategies address sodium within a broader nutritional context, such as front-of-pack labeling, marketing restrictions, and food procurement policies for public institutions. Taxation of high-sodium foods could address sodium specifically or be addressed through a broader intervention such as a “junk food” tax. When deciding on the specific strategies to include, countries may also take into consideration whether there is political will to implement sodium-specific strategies or if taking a broader approach to policies for a healthy diet will be more acceptable. In many cases it may be both more acceptable as well as more effective to take a broader healthy diet approach with a strong emphasis on sodium reduction.

Only a small number of strategies for population sodium reduction have been rigorously evaluated on a national scale. Two industry-facing strategies, front-of-pack labeling and food reformulation targets, have been widely studied, with evidence from multiple countries on reduction of population sodium intake. Other interventions, including taxation and mass media campaigns, have been used and evaluated widely for other public health problems such as tobacco use, with more limited evidence in terms of sodium. Two strategies, regulation of marketing food and beverages to children as well as public food procurement policies, have a smaller but growing body of evidence, suggesting that they can be effective strategies for sodium reduction in the context of improving overall dietary quality. Because existing strategies for sodium consumed at home have been primarily educational in nature and often not scalable, we specifically sought further strategies that could fill this gap. We determined subsidization and/or market scale up of low-sodium salt meet our criteria. Although this strategy does not have the large-scale evidence as some other strategies, pilot studies have suggested this may be an effective strategy for a scalable population sodium reduction approach.

### Areas for Future Research

Key research questions that will further strengthen the evidence base for the seven recommended strategies remain unanswered. Among these gaps is a need to identify the sodium-specific outcomes of nutrient-comprehensive strategies, such as front-of-pack labeling, marketing regulations, or public food procurement policies. Although these three interventions have some suggestive evidence of effectiveness, understanding the impact of these interventions specific to reduced sodium consumption remains an important question to be answered. Another gap is the lack of an evaluation of a large-scale intervention for low-sodium salt subsidies, including interventions that ensure sustainable uptake of these salt substitutes. Suggestive studies from China and Peru are a good start, but there has yet to be a large-scale government intervention implemented.

Another consideration for future monitoring activities and impact evaluations is to measure all primary nutrients of concern in order to avoid unintended consequences. Even if sodium reduction is the targeted nutrient for a policy, the strategy should not negatively influence consumption of other nutrients such as sugar or unhealthy fats. For example, manufacturers implementing a policy mandating sodium reformulation targets should not meet the targets by increasing the density of sugar. Iodine intake should be simultaneously monitored to ensure adequate levels, and depending on the outcomes, adjusting iodine fortification efforts may be required. This could include promoting purchase and use of iodized salt among households, increasing iodine levels during manufacturing of salt, or increasing use of iodized salt among packaged food producers and restaurants. Additionally, evaluations should assess any impact on health disparities and considerations should be given to how strategies can be structured to reduce disparities in the population.

One of the primary areas for future work is to build up the base of high-impact strategies for sodium consumed from food prepared outside the home. Further work is required to develop and evaluate solutions, especially for restaurants, street food vendors, canteens, and supermarkets. Restaurants are a particularly important environment for innovation, as a variety of countries are seeing large and/or a growing consumption of restaurant food. Although some scalable restaurant strategies have been implemented among chain restaurants, including menu warning labels on high sodium dishes or setting voluntary targets for restaurants, these have not been well evaluated, and it is not known whether they have significant benefit [115,116]. Likewise, supermarkets could be utilized to encourage and/or educate consumers on low-sodium options, but little evidence exists as to whether this has any effect. Innovation in the restaurant sector is particularly important in order to identify scalable, effective, and large benefit strategies.

In addition to building the evidence base for out-of-home settings, continued innovation for reducing sodium consumed in the home is important for many countries. A growing number of potentially promising approaches have been proposed, but these need more thorough testing in the field to understand whether they are scalable and will have a population rather than individual benefit. For example, researchers in Japan proposed a device to self-monitor urinary sodium-to-potassium ratio that might encourage behavior change among individuals [117]. Researchers in China have developed a smartphone application (AppSalt) to reduce sodium intake among schoolchildren and their families through a combination of education, salt monitoring, decision support, reminders, and supportive modules [118]. It is especially important for strategies that take an individual approach to not only aim to educate the individual, but to incorporate methods that will go beyond to motivate and/or incentivize the individual to make the desired behavior changes.

## 5. Conclusions

Scalable strategies supported by evidence of effectiveness and innovation exist to reduce population sodium intake, particularly strategies that target the packaged food sector. Governments have an opportunity to develop a multi-component package to reduce sodium and improve the health of the population by preventing morbidity and mortality from diet-related non-communicable disease. As many of the recommended strategies target nutrients in addition to sodium, there is potential for an even broader health impact from reduction in other nutrients such as trans fat or sugar and from promotion of fruit and vegetable consumption. Investment in a combination of these seven strategies offer a high-impact, cost-effective method to reach the global target of a 30% relative reduction in mean population sodium intake by 2025.

## Figures and Tables

**Figure 1 nutrients-12-02543-f001:**
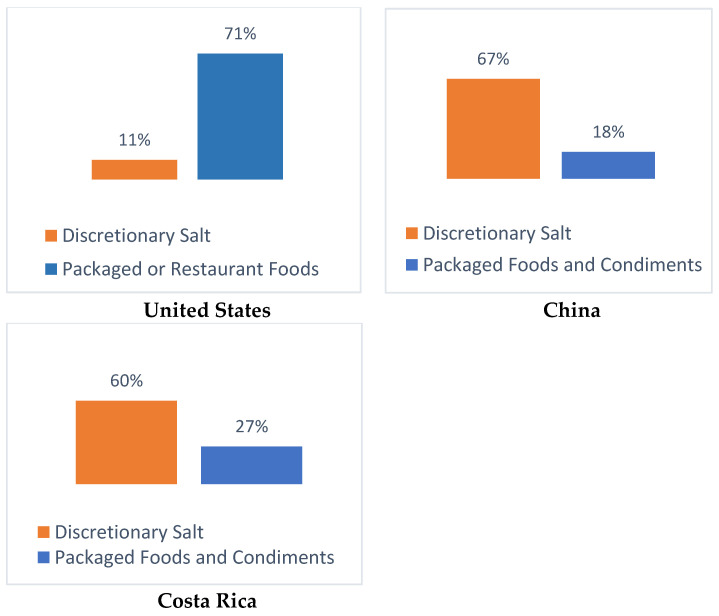
Primary Sources of Sodium from the United States [8], China [9], and Costa Rica [10].

**Table 1 nutrients-12-02543-t001:** Comprehensive salt reduction program in United Kingdom and South Korea.

United Kingdom	South Korea
**Start date**: 2003/2004	**Start date**: 2012
**Target**: Reduce the population average salt intake to 6 g/day	**Target**: Reduce population sodium consumption by 20%, to 3900 mg/day, by 2020
**Package components:**	**Package components:**
Surveillance: salt intake and salt levels in foods	Consumer awareness campaign
Reductions in processed food: voluntary salt targets with timelines, strict monitoring, and threat of legislation	Increased availability of low-sodium foods at school and worksite meal services
Reductions in restaurants	Increased availability of low sodium meals in restaurants
Clear nutrition labeling	Voluntary reformulation of processed foods to lower the sodium content
Public health awareness campaign	Development of low-sodium recipes for food prepared at home
**Results:**	**Results:**
Salt intake decreased by 1.4 g per day, or 15%, between 2003/2004 to 2011	23.7% reduction in population sodium intake
From 2003 to 2011, mean blood pressure reduced by 3.0/1.4 mmHg, as well as decreased mortality from stroke by 42% and ischemic heart disease by 40%	Significant decreases in population blood pressure and in hypertension prevalence between 2010 and 2014 in adults above 30 years (men from 33.5% to 26.0%, and women from 25.2% to 21.7)

**Table 2 nutrients-12-02543-t002:** Existing sodium reduction strategies.

**Sodium from packaged foods**
Labeling: front-of-pack labeling regulations
Labeling: mandatory nutrient declaration on labels
Labeling: regulating nutrition/health claims on food packaging
Food reformulation targets for packaged food (voluntary or mandatory)
Regulation of marketing of foods and nonalcoholic beverages to children
Fiscal policies: taxation on high sodium foods
Supermarket interventions using product, placement, price, or promotion strategies (4Ps)
**Sodium from food prepared outside the home**
Standards for sodium as part of food procurement policies for public institutions
Restaurants: menu labeling of high or low sodium items (primarily chain restaurants)
Restaurants: removal of salt shakers and high sodium condiments from tables
Restaurants: chef training on reducing sodium in food
Restaurants: requiring the provision of low sodium or no-sodium added items on menus
Restaurants: food reformulation targets for restaurants (voluntary or mandatory; primarily chain restaurants)
**Sodium added in the home**
Mass media campaigns
Community education (e.g., through schools, community groups, workplaces, etc.)
Individual education and counselling (usually through primary health care)
Increase uptake of low sodium salt (promotion, distribution, subsidies)

**Table 3 nutrients-12-02543-t003:** High-impact strategies meeting the framework criteria.

	Scalable and Sustainable	Evidence of Effectiveness or Innovation	Large Benefit
**Sodium consumed from packaged foods**			
Front-of-pack labeling regulations	✓	Suggestive evidence of effectiveness	Modeling studies available
			Theoretically large benefit
Food reformulation targets for packaged food (voluntary or mandatory)	✓	Rigorously evaluated	Impact evaluations conducted on sodium and health outcomes
Regulation of marketing of foods and nonalcoholic beverages to children	✓	Suggestive evidence of effectiveness	Theoretically large benefit
Fiscal policies: taxation on high sodium foods	✓	Suggestive evidence for sodium, rigorously evaluated for other topic areas (e.g., sugar sweetened beverages)	Modeling studies available
			Theoretically large benefit
**Sodium consumed from food prepared outside the home**			
Standards for sodium as part of food procurement policies for public institutions	✓	Suggestive evidence of effectiveness	Modeling studies available
			Theoretically large benefit
**Sodium consumed at home**			
Mass media campaigns	✓ ^1^	Some suggestive evidence of effectiveness, rigorously evaluated for other topic areas (e.g., tobacco)	Theoretically large benefit
Increase uptake of low-sodium salt (promotion, distribution, subsidies)	✓	Innovative approach for sodium added at home	Modeling studies available	
			Theoretically large benefit

^1^ May not be sustainable due to recurrent costs for on-going or repeat campaigns.

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
