# Peer review of "Priority Actions to Advance Population Sodium Reduction"

_nutrients, 2020, doi:10.3390/nu12092543_

Round 1
Reviewer 1 Report
Ide et al. had reviewed the approaches for reducing sodium intake worldwide in this manuscript. I have several comments for this manuscript.
1. The reviews on sodium reduction from population approach have been discussed in many previous articles and the conclusions/ key messages tend to be similar. Thus, this article seems not adding new perspectives and/or findings compared to others. Authors pointed out "innovative approach for sodium added at home" in Table 3; however, it remains ambiguous. I would suggest adding new points from individual approach (e.g. see discussion on self-monitoring device using Na/K ratio in PMID:28678188, 29093302, 30996260) and speculate better approach than the conventional papers for the countermeasure which may work in practice. There are too many ordinary papers telling us that high sodium intake is a serious problem, although few practical countermeasures had been addressed to deal this problem.
2. Conventional population approach seems working; however far away from our expectations. Non-communicable diseases related to sodium reduction are mostly "silent killer" diseases (e.g. hypertension). Therefore, it might be difficult to raise the awareness of it, make people motivated to be involved and design incentive to make multiple behavior change/ education approaches work. Speculations on these point is apparently lacking in this manuscript. How can we combine population and individual approaches to make it really work?
Author Response
We thank Reviewer 1 for the insightful comments and suggestions. We have incorporated the following changes:
Point 1. The reviews on sodium reduction from population approach have been discussed in many previous articles and the conclusions/ key messages tend to be similar. Thus, this article seems not adding new perspectives and/or findings compared to others.
Author response:
We acknowledge that this is not the first article to review population approaches for sodium reduction. As explained in lines 83-87, existing review articles on this topic (Hyseni et al. and McLaren et al.) conclude that multi-component approaches that include population-level policy interventions tend to have the largest impact. While we think these conclusions are helpful, we believe this paper goes further, as Reviewer 2 notes, in helping countries determine which strategies to include in a multi-component intervention. For the many countries (especially in Asia and most lower-income countries) that consume a majority of their sodium from discretionary sources (at home during cooking or at the table), guidance around how to reduce sodium in the primary source of sodium is still needed. For these countries, policy interventions that change the food supply or external environment will not sufficiently address the sodium consumed in the home. Thus, the existing frameworks which advocate for multi-component policy interventions may not always be sufficient or fully appropriate for all countries. We believe a more nuanced framework to assess interventions is required (see explanation in lines 117-121). Our framework addresses this gap in the existing frameworks by focusing on scalable and sustainable interventions that have evidence of effectiveness or innovation and can achieve a large population benefit. We particularly wanted to create room in the framework to allow for innovative strategies that might address areas that do not have many proven strategies (or obvious policy approaches), such as for discretionary salt sources.
To address this, we have added text to the introduction to highlight the fact that the current manuscript builds on previous work, but specifically provides countries and policy makers with a framework to help identify priority interventions. We have also highlighted the current lack of stand-alone individual-level interventions that meet the criteria described in our framework in the discussion.
For specific revisions, see:
Lines 87-90. Added: “The framework we apply here builds on published reviews and experience during implementation to provide additional guidance for governments and others to identify priority interventions and take systematic action to reduce sodium intake.”
Lines 445-452: Added a description of the lack of stand-alone individual-level interventions that meet our framework: “We did not include community education or individual education and counseling as recommended standalone strategies, as these interventions rely on individual-level behavior change and are therefore difficult to scale and achieve population level benefits. These education-based strategies are particularly challenging for sodium reduction, as there are typically no visible consequences to a high-sodium diet and sodium-related non-communicable diseases often have no symptoms until they have reached an advanced stage. Thus, many individuals lack the motivation to change, regardless of their level of awareness.”
Point 2. Authors pointed out "innovative approach for sodium added at home" in Table 3; however, it remains ambiguous. I would suggest adding new points from individual approach (e.g. see discussion on self-monitoring device using Na/K ratio in PMID:28678188, 29093302, 30996260) and speculate better approach than the conventional papers for the countermeasure which may work in practice. There are too many ordinary papers telling us that high sodium intake is a serious problem, although few practical countermeasures had been addressed to deal this problem.
Author response:
Strategies included in Table 3 represent strategies that meet the proposed framework from Box 1. We agree that there are a number of new, interesting, and innovative approaches starting to appear in the literature; however, we did not choose to include all of these in Table 3 as it is not clear that these approaches meet all three of the framework criteria. For example, it is not clear whether the self-monitoring device mentioned by the reviewer is a scalable strategy for many populations due to the cost of the device, the need for individual education on how to use the device, and the uncertainty on whether uptake would be very high. Conclusions from the literature do not yet support a large benefit. We do recognize the importance of these types of innovative approaches and agree with the reviewer that more emphasis on these “practical countermeasures” should be given. Thus, we have added this point in the discussion in the ‘areas for future research’ section along with examples.
For specific revisions, see:
Lines 513-523 added: inserted a paragraph describing the potential of new and innovative strategies to reduce sodium consumed in the home, with the caveat that these interventions need to be further tested in the field to understand whether they are scalable and will have a large population benefit. The self-monitoring device suggested by the reviewer and another study involving a salt reduction mobile phone app were added to this paragraph.
Point 3. Conventional population approach seems working; however far away from our expectations. Non-communicable diseases related to sodium reduction are mostly "silent killer" diseases (e.g. hypertension). Therefore, it might be difficult to raise the awareness of it, make people motivated to be involved and design incentive to make multiple behavior change/ education approaches work. Speculations on these point is apparently lacking in this manuscript. How can we combine population and individual approaches to make it really work?
Author response:
In the original submission, we included a brief discussion on this topic in the second paragraph of the discussion. We fully agree with the reviewer regarding the difficulty of relying on educational approaches alone and the need to combine with population approaches. While we had addressed this somewhat in the discussion, we agree that this point can be made stronger and have added a few sentences to this paragraph as well as a brief sentence on taking a motivational/incentivization approach in the newly added paragraph in the ‘Areas for future research’ section.
We believe the final question in the reviewers comments on this point (How can we combine population and individual approaches to make it really work?) has been addressed in lines 452-456 by providing examples of how educational approaches can be combined with population approaches in order to improve their effectiveness: “Education can be used to complement many of the included strategies and improve their chances of achieving large population benefits. For example, the effectiveness of front-of-pack labeling policies can be strengthened through a complementary education campaign on how to use/interpret the new labels. Similarly, health care workers could be trained to promote low-sodium salt to their patients.”
For specific revisions, see:
- Lines 445-453: incorporated the suggested concepts of acknowledging the challenges of educational/behavior change approaches and describing how education approaches can complement population approaches.
- Lines 521-523: Incorporated a sentence encouraging educational approaches to take an approach that motivates and/or incentivizes the individual towards behavior change.
Reviewer 2 Report
This is an updated and well-done paper. I agree absolutely with the ideas expressed in the paper.
I have only one suggestion to the authors: everywhere we deal with sodium restriction, iodine intake should be taken into account.
My country is fighting from at least a century (from 1924) against iodine deficiency. I believe that parallel iodine and sodium intake have importance.
So, the only minor revision I suggest is to include at least a sentence in the abstract and a short paragraph in the main text dealing with the problem of possible iodine deficiency arising from sodium restriciton and the way to avoid this, taking into account the three primary sources of dietary sodium (as correctly reported in the text).
Author Response
We thank Reviewer 2 for the insightful comments and suggestions. We have incorporated the following changes:
Point 1. This is an updated and well-done paper. I agree absolutely with the ideas expressed in the paper. The only minor revision I suggest is to include at least a sentence in the abstract and a short paragraph in the main text dealing with the problem of possible iodine deficiency arising from sodium restriction and the way to avoid this, taking into account the three primary sources of dietary sodium (as correctly reported in the text).
Author response:
We thank the reviewer for this important feedback and agree that a discussion on the very related issue of salt iodization was lacking. As suggested, we have incorporated this issue into the abstract, added a paragraph addressing the issue in the introduction, and mentioned potential solutions in the discussion.
For specific revisions, see:
- Lines 11-12 in the introduction to briefly highlight the issue
- Lines 70-76: a paragraph was added to the introduction to explain the potential concern of inadequate iodine intake due to salt reduction and the need to harmonize salt reduction and iodine intake programs. Two references added on the topic (Campbell et al. and Charton et al.).
- Lines 496-499: iodine intake mentioned in the discussion to highlight need for monitoring of iodine intake and the potential strategies to increase levels of iodine fortification and intake among the population.
Round 2
Reviewer 1 Report
The authors added statements in lines 446-452 that they avoided to include individual approach since it relies on individual-level behavior change. However, conventional population approaches are working but still gaps between the actual and targeted sodium intake levels remain significant. Therefore the reasoning raised by the authors (justification for focusing on population approach only) is not convincing and nothing new for this topic. The authors also stated that the education-based strategies are particularly challenging for sodium reduction; however, these are common among NCDs (lifestyle related diseases). NCDs is a silent killer diseases and various approaches had been proposed. The authors need to provide updates (e.g. highlight individual approach and/or other approaches) and speculate how we can mitigate the gap between the actual and targeted sodium intake level by enhancing the motivation and raise awareness in order to change the current situation (the authors may reference approaches taken for NCDs and speculate how we can implement for sodium intake reduction).
Author Response
We thank the reviewer for the suggested changes. We agree that the current draft has not provided a full review of sodium reduction strategies that target an individual approach. We have reviewed some successful examples in the discussion. However, we have also concluded that although these examples were successful at lowering the participants’ sodium intake, there are major limitations to scaling these interventions. After reviewing existing research on sodium reduction interventions, we have concluded that the best way to achieve change on an individual level is by making structural shifts, making the default choice the healthier choice. Individual and collective changes can be synergistic.
Regarding the issue of referencing approaches which have been taken for NCDs, we find that many of the same limitations found for sodium reduction individual approaches are similar for other NCD areas. Clark’s 2014 article, Medicalization of global health 3: the medicalization of the non-communicable diseases agenda, describes well the limitations of the individual approach explaining, “most recommended strategies tend towards the individualistic approach and do not address root causes of the NCD problem”. “(They) are often actually expectations of individual and behavioural change, which will have limited success and impact, and deflect attention away from government policies or regulation of industry”. The article explains these approaches medicalize the framing of a health problem, without considering the broader social and environmental contexts. Frieden’s 2011, A Framework for Public Health Action: The Health Impact Pyramid, similarly demonstrates how interventions designed to help individuals rather than entire populations could theoretically have a large population impact, but would require being universally and effectively applied. Interventions taking a broader approach to address socioeconomic determinants or public health interventions that change the context for health require less individual effort and have the greatest population impact.
For specific revisions, see:
Lines 356-359: Added: “Some interventions reviewed that address home consumption of sodium through other individual or behavior change approaches were found to successfully reduce sodium consumed at an individual level. However, these strategies have not been included as they…”
Line 447: Added: “at a population scale”
Lines 448-462. Added: “Studies and pilot interventions that relied primarily on an educational/individual approach have reduced sodium consumption among study participants. For example, the Trials of Hypertension Prevention (TOHP) included comprehensive education and counselling on sodium reduction among prehypertensive adults in the United States. Results showed that participants in the intervention achieved significant reductions in sodium and a 25-30% lower risk of cardiovascular outcomes in the 10 to 15 years following the trial. [110] Another study from northern China involving a school-based salt reduction education program (School-EduSalt) found reduced salt intake and blood pressure among children and their family members in the intervention group.[111] Such examples demonstrate that with high levels of investment, individual-focused educational interventions can successfully increase motivation and knowledge leading to sodium reduction behaviors. However, these investments are likely cost-prohibitive to conduct at scale and difficult to sustain without a change in the food environment. A review of such educational studies warns that despite the possibility of success, they are unlikely to be adequate in reducing population salt intake to the recommended levels and would benefit from being implemented alongside more structural interventions. [112]
Therefore, due to the difficulty of scaling these interventions and achieving population level benefits, we did not include…”
- Cook, N.R.; Cutler, J.A.; Obarzanek, E.; Buring, J.E.; Rexrode, K.M.; Kumanyika, S.K.; Appel, L.J.; Whelton, P.K. Long term effects of dietary sodium reduction on cardiovascular disease outcomes: observational follow-up of the trials of hypertension prevention (TOHP). Bmj 2007, 334, 885-888, doi:10.1136/bmj.39147.604896.55.
- He, F.J.; Wu, Y.; Feng, X.X.; Ma, J.; Ma, Y.; Wang, H.; Zhang, J.; Yuan, J.; Lin, C.P.; Nowson, C., et al. School based education programme to reduce salt intake in children and their families (School-EduSalt): cluster randomised controlled trial. Bmj 2015, 350, h770, doi:10.1136/bmj.h770.
- Trieu, K.; McMahon, E.; Santos, J.A.; Bauman, A.; Jolly, K.A.; Bolam, B.; Webster, J. Review of behaviour change interventions to reduce population salt intake. Int J Behav Nutr Phys Act 2017, 14, 17, doi:10.1186/s12966-017-0467-1.